# Survey of Flemish Poultry Farmers on How Birds Fit for Transport to the Slaughterhouse Are Selected, Caught, and Crated and Their Opinions Regarding the Pre-Transport Process

**DOI:** 10.3390/ani14223241

**Published:** 2024-11-12

**Authors:** Femke Delanglez, Anneleen Watteyn, Bart Ampe, An Garmyn, Evelyne Delezie, Gunther Antonissen, Nathalie Sleeckx, Ine Kempen, Niels Demaître, Hilde Van Meirhaeghe, Frank André Maurice Tuyttens

**Affiliations:** 1Animal Sciences Unit, Flanders Research Institute for Agriculture, Fisheries and Food (ILVO), 9090 Melle, Belgium; femke.delanglez@ilvo.vlaanderen.be (F.D.); anneleen.watteyn@ilvo.vlaanderen.be (A.W.); bart.ampe@ilvo.vlaanderen.be (B.A.); evelyne.delezie@ilvo.vlaanderen.be (E.D.); 2Department of Veterinary and Biosciences, Faculty of Veterinary Medicine, Ghent University, Heidestraat 19, 9820 Merelbeke, Belgium; 3Department of Pathobiology, Pharmacology and Zoological Medicine, Faculty of Veterinary Medicine, Ghent University, 9820 Merelbeke, Belgium; gunther.antonissen@ugent.be; 4Experimental Poultry Centre, 2440 Geel, Belgium; nathalie.sleeckx@provincieantwerpen.be (N.S.); ine.kempen@provincieantwerpen.be (I.K.); niels.demaitre@provincieantwerpen.be (N.D.); 5Vetworks, 9880 Aalter, Belgium; hilde.vanmeirhaeghe@vetworks.eu

**Keywords:** fitness for transport, industry, transport system, poultry, handling

## Abstract

Poultry management prior to transport can result in animal stress, injuries, and mortality. Here, we aimed to gather information from Flemish poultry farmers about their current pre-transport practices (i.e., selection of unfit chickens, catching preparation, catching, and crating) for spent hens and broilers. The results showed that a minority of farmers performed catch preparation, such as pre-selecting chickens unfit for transport. Practices on layer farms were less aligned with the EU legislation for water and feed withdrawal than on broiler farms. All birds were caught inverted except for one broiler farmer who used mechanical catching. Although mechanical catching may involve extra costs, increased biosecurity risks, and specific recommendations for the stable (height and width), it was reported as the preferred method for broiler catchers’ well-being. Upright catching was considered better for animal welfare than catching more than three chickens by one/two legs, mechanically, or by the wings. Poultry farmers should be sensitized about the need for pre-catch selection, including clear guidelines about judging which birds are fit for transport. Pre-catch measures (e.g., closing the area under the aviary system, removing litter) can streamline the catching process and reduce animal suffering.

## 1. Introduction

At the end of the production phase, broilers and laying hens are usually manually caught and loaded into containers or crates by a group of catchers and then transported to the slaughterhouse [1,2]. In some countries where farms are located far away from slaughterhouses (e.g., in Canada, the US, and Finland), spent layers are gassed on farms and are not used for consumption as they have a lower economic value than broilers [3,4,5].

This paper focuses on the first phase of the pre-slaughter stage prior to the transport of broilers and spent laying hens, which includes selecting birds not fit for transport, other preparations before the catching event, and the catching, crating, and loading of the birds. The pre-transport phase represents an important risk for potential animal welfare problems such as thermal discomfort, bone lesions (e.g., fractures and dislocations), skin lesions (e.g., subcutaneous hemorrhages), handling stress, restriction of movement, fatigue, fear, and mortality [1,6,7].

EU regulations concerning birds’ fitness for transport [8] stipulate that “no animal shall be transported unless it is fit for the intended journey, and all animals shall be transported in conditions guaranteed not to cause them injury or unnecessary suffering”. Birds not fit for transport should receive appropriate treatment or be immediately culled [8] as this helps to ensure food safety and reduce economic losses [9,10]. Given the challenge of properly checking birds’ fitness for transport during catching and crating, evaluating the birds’ fitness for transport should be performed as close as possible to the catching event [11]. This fit-for-transport inspection should be carried out in addition to the recommended routine flock inspections (at least twice daily for broilers [12] and at least once a day for laying hens [13]) throughout the production cycle. In Belgium, Belplume promotes uniform standards for animal welfare to promote adequate animal care [14].

In addition to this special selection of birds just before the catching event, other preparations are necessary, such as litter removal (for laying hens only), feed and water withdrawal, and lighting adjustments. Specifically, in non-cage systems of laying hens, litter in the alleys should be removed to allow containers and crates inside the stable as close as possible to the hens [7]. Feed should be withdrawn at the correct time (8–12 h before slaughter) to maximize the chance that the gastrointestinal tract is empty at the time of slaughter with the aim of preventing carcass contamination. This timing is the best balance between minimizing carcass rejection and minimizing the risk of starvation prior to slaughter [15]. To maximize gastrointestinal evacuation, four hours of light is recommended after feed withdrawal [16]. Unduly long periods of feed withdrawal and light adjustment can cause significant stress, which is associated with negative effects on meat quality and, in turn, can result in economic losses [16]. Furthermore, water should remain available to the birds immediately prior to the catching event; this not only keeps the birds free from thirst but also helps to clear their gastrointestinal tract [16].

The actual catching and loading of the birds can cause fear, stress, fractures, and lesions. This results in animal suffering as well as significant economic losses for producers and slaughterhouses due to carcass rejections [17,18]. Laying hens and broilers require different approaches to catching and loading. Laying hens in the EU are caught and loaded at 65–100 weeks of age [7,19], fast-growing broilers at 6–7 weeks, and slow-growing broilers at 7–12 weeks [16]. In the EU, layers are housed in either enriched cages or non-cage systems (floor housing and aviary systems) [1], while broilers are usually kept in floor housing systems [20]. Laying hens are always caught manually, whereas broilers can also be caught mechanically [1,2,21]. In the case of manual catching, both laying hens and broilers are usually caught by one or both legs (despite being prohibited in the EU [8]) and carried upside-down. Upright catching, where the birds are supported under the abdomen and the wings are kept against the body, has been recommended as a more animal-friendly approach (less agitation, stress, and injury) [7,18,22].

In broiler farming only, mechanical catching is sometimes used. Studies that have compared animal welfare during mechanical versus manual catching are not unanimous. One study [23] reported improved animal welfare during mechanical catching, while [24] concluded the opposite. Heterogeneity between study outcomes likely relates to differences in the type of mechanical harvester (e.g., a sweeping system with rotating rubber fingers or conveyor belts that transport the chickens into the containers) and the experience and attitude of the staff operating the machine. If mechanical catching is performed correctly, it can reduce the risk of animal injuries compared to manual catching [7,25].

After catching, chickens are crated in either containers (where drawers must be opened and closed during filling) or crates (the chickens are placed in the crate through an opening at the top). The crating of broilers is easier in containers (wide drawers) than in crates (smaller openings). The use of crates can result in wing injuries, but containers with wide drawers are associated with a higher risk of broiler escape [26]. For laying hens, crates are associated with a higher prevalence of body part entrapments (0.006 ± 0.021% vs. 0.037 ± 0.041%) but an overall reduced risk of injuries (0.014 ± 0.020% vs. 0.002 ± 0.040%) in comparison with containers [27].

Previous research has studied selection for fitness for transport, catching, and crating, but little is known about the poultry farmers’ perspectives. In the present study, Flemish poultry farmers were surveyed to obtain an in-depth view of the entire pre-transport process on commercial farms. As poultry farmers are responsible for the chickens during the entire pre-transport phase, their opinion is essential regarding selecting poultry unfit for transport, catching, and crating. Their insights can be decisive regarding the successful implementation of new or adapted practices.

## 2. Materials and Methods

A quantitative survey about the selection, catching, and crating process (laying hens and broilers) was conducted with poultry farmers in Flanders (Belgium). The survey was based on prior qualitative face-to-face interviews with farmers (*n* = 5) based on open questions about the procedures and farmer opinions on selecting, catching, and crating poultry. The answers to the open questions were used to formulate appropriate and realistic questions and possible answers for the quantitative survey. The surveys for laying hen and broiler farmers were created with an online survey tool (LimeSurvey, www.LimeSurvey.org (accessed on 13 April 2022).

### 2.1. Respondents

The contact details (phone number, address, and email address) of 156 laying hen and 203 broiler farmers were obtained from the Flanders Department of Agriculture and Fisheries. To the best of our knowledge, this group represented a random sample of the total population of 177 laying hen farmers and 522 broiler farmers registered in Flanders in 2022 [28]. The farm selection criteria were (1) a minimum number of 200 birds and (2) marked/registered as a poultry farm. In April 2022, the poultry farmers on the contact list were invited by email to participate. Follow-up calls were made at the beginning of June 2022 to increase the response rate, and the survey was closed at the end of June 2022. In total, 31 laying hen (20%) and 48 broiler (24%) farmers filled out the questionnaire completely, and another 134 farmers (49 laying hen and 85 broiler farmers) filled it out partially. Depending on the question, the number of responses varied between 84 and 133 for broiler farmers and between 42 and 80 for laying hen farmers.

### 2.2. Surveys

Two surveys were administered: one for laying hen farmers and one for broiler farmers. Most questions were similar on both surveys; only the answer options for some questions differed somewhat (see Appendix A). Both surveys were divided into three sections. The first section gathered demographic data about the farmer respondent (age, gender, and highest level of education completed) and general information about the farm, including the location, poultry type, slaughter weight (kg), and slaughter age (in days for broilers, and weeks for laying hens) of the most recently completed production cycle. In the second section, multiple choice questions were asked about the procedure for selecting unfit chickens to be culled during the most recent inspection rounds during the production cycle (individual performing the culling, reasons for culling, bottlenecks, and advantages). Additional questions were asked regarding the most recent extra selection before catching to remove birds unfit for transport.

In the third section, farmers were asked to focus on the most recent catching and loading event on their farm and to provide details on (1) the preparations they had made, including litter removal (laying hens), closing of laying nests (laying hens), changes to the light schedule, and withdrawal of food and water (multiple choice); (2) the timing of the catching event (start and end times); (3) the used catching method, the number of chickens per catching method, the number of catchers, and type of container/crate, and its influence on animal welfare; (4) the presence and role of the poultry farmer; points of attention during catching and loading (multiple choice); (5) the physical and mental strain for the catchers (on a scale from 0 to 100%, with 0% = not present at all and 100% = present); (6) bottlenecks (e.g., noise, stress, injuries, and inefficiency linked to the catching team’s behavior) (on a scale from 0 to 100%, with 0% = not present at all and 100% = present); (7) communication with the catching team; and (8) economic data (price per chicken and the method of payment for the catching team).

In addition, they were asked to rank (from most preferred to least preferred) nine (laying hens) or ten (broilers) different catching methods with regard to economy and time efficiency, the well-being of the catcher, and animal welfare. The catching methods were (a) two chickens in one hand by one leg, (b) two chickens in one hand by two legs, (c) three chickens in one hand by one leg, (d) three chickens in one hand by two legs, (e) more than three chickens in one hand by one leg, (f) more than three chickens in one hand by two legs, (g) one chicken upright, (h) two chickens upright, (i) mechanical harvesting (only for broilers), and (j) wing catching. Finally, the participants could fill in their email addresses and confirm whether they were interested in being informed about the survey results and the larger research project on the pre-slaughter phase in poultry.

### 2.3. Statistical Analysis

The response data were analyzed using R (version 4.2.1). First, the results are presented descriptively and in some cases, a statistical analysis (e.g., a linear and logistic regression) was performed. Continuous variables (e.g., age of the farmer, number of chickens per catching movement, the number of catchers, the influence of the type of container/crate on animal welfare, the physical and mental strain for the catchers, communication with the catching team, price per chicken, and opinion of different catching methods) were analyzed using linear regression, with the type of livestock farmer (laying hen or broiler) as a fixed effect. Differences were reported as least squares mean. Continuous data were assumed to be normally distributed based on a visual inspection of the model residuals (graphical assessment using QQ-plot and histograms). Binomial variables for selecting chickens (present or absent; e.g., location farm, performer, reasons for culling, bottlenecks, advantages, and extra selection) and catching and loading (present or absent; e.g., preparations, for or against daytime loading, which catching method, presence, and role of the poultry farmer, points of attention, bottlenecks, method of payment of the catching team) were analyzed using logistic regression, with the type of livestock farmer as a fixed effect. Differences were reported as the back-transformed least squares mean (the proportions or percentages of occurrence). In the case of different answer options, the answer option and the interaction with the type of livestock farmer were included in the model. Within each farmer type, a post hoc test was used to test the difference between the different answer options, and Tukey’s correction was performed. The answers regarding the type of livestock farmer were compared, as well as between the types of livestock farmers (laying hen vs. broiler). Additionally, *p*-values less than 0.05 were considered significant.

## 3. Results and Discussion

### 3.1. General Information

Most laying hen and broiler farmers were older than 45 years. The average age was 53 years (layers) and 48 years (broilers) (*p* = 0.004), and the majority were male with a high school diploma as their highest degree (Table 1). Most layer and broiler farms were located in Antwerp and West-Flanders in comparison to Flemish Brabant (*p* < 0.05 and *p* < 0.0001) and East-Flanders (*p* < 0.05) and Limburg (only broilers) (*p* < 0.001) (Table 1). Most laying hen farmers used an aviary (42%) or an enriched cage (37%) system. The number of chickens per stable typically ranged from 10,000 to 30,000 (Table 1). The mean slaughter age and weight of broilers was 28 ± 13 days and 1.70 ± 0.73 kg at the time of thinning and 38 ± 14 days and 2.44 ± 0.76 kg at the end of the production cycle (*n* = 118). The average slaughter age of the white laying hens was 90 ± 20 weeks (*n* = 11), and for the brown laying hens, 85 ± 18 weeks (*n* = 41). In general, white layers are more fearful (longer duration of tonic immobility and higher corticosterone levels) than brown layers, including when handled by humans (more flighty) [29]. 

### 3.2. Selection of Unfit Chickens During the Production Cycle

As a part of the daily routine inspections throughout the production round, the selection of (moribund) chickens to be removed from the flock was predominantly done by the poultry farmer personally and on a minority of farms by farm staff (Table 2). Employing staff to select chickens was more common for layers than broilers (*p* = 0.03). In contrast, the farmer participated more on broiler farms than on laying hen farms (*p* = 0.04). Furthermore, on both laying hen and broiler farms, the farmer selected chickens the most when compared to the farm staff and the company’s veterinarian (*p* < 0.0001). In general, more laying hens than broiler farmers selected chickens that were ill or injured (24% vs. 6%; *p* = 0.004). During the production cycle, birds could be ill or injured [30].

For layers, the most commonly reported reasons for removal were leg problems/lameness (57%), followed by being feather-pecked (43%), and *E. coli* infection (32%). Stunted growth and huddling were less commonly reported as a reason for culling. Feather pecking can potentially lead to cannibalism and high mortality rates [31]. EU legislation for laying hens does not specify criteria for selecting unfit chickens during the production cycle [13], which hinders laying hen farmers from knowing whether they comply with legislation. We, therefore, recommend clearer EU guidelines for laying hen farmers according to the criteria for selecting unfit laying hens during the production cycle.

For broilers, leg problems/lameness (85%) and stunted growth (71%) were more commonly mentioned as reasons for culling during the production cycle compared to feather pecking (3%), huddled animals (13%), and *E. coli* (26%) (*p* < 0.0001) (Table 2). Lameness (86.4%) and broken legs (79.5%) are also noted as culling reasons in the literature [32]. The EU legislation for broilers stipulates criteria for selecting unfit chickens during the production cycle, e.g., seriously injured animals or evident signs of health disorders, including walking difficulties, severe ascites, or severe malformations [12]. Neither severe ascites nor severe malformations were indicated by the surveyed broiler farmers. Although these criteria were not present as options in the survey, a text box was provided so that poultry farmers could have mentioned either of these or other criteria.

Laying hen farmers reported time constraints as the primary bottleneck for selection more often than financial reasons (*p* = 0.04) or insufficient knowledge (*p* = 0.002). In contrast, broiler farmers reported financial reasons and time constraints more often than insufficient knowledge (*p* < 0.0001). Selecting unfit birds is time-consuming but saves costs by reducing feed costs and disease spread [33]. Both groups rarely mentioned that a lack of knowledge about which poultry to select represented a bottleneck (Table 2). The farmers should focus on the procedure of selecting unfit birds, while veterinarians should advise farmers/farm staff on how to identify unfit birds [34,35].Concerning the advantages of selection, laying hen farmers prioritized animal welfare over flock uniformity and reduced feed waste (*p* = 0.03). Broiler farmers prioritized animal welfare (*p* = 0.0005) and uniformity (*p* = 0.03) over reducing feed waste. Overall, broiler farmers considered animal welfare, flock uniformity, and reducing feed waste to be more important than laying hen farmers (Table 2). For both broiler and laying hen farmers, it is essential to have an efficient and fast inspection protocol based on scientific and legislative recommendations.

### 3.3. Preparations Before Catching and Loading

Figure 1 provides an overview of the average timeline of the preparatory phase before the onset of the catching and loading of birds, as reported by the laying hen (*n* = 51) and broiler farmers (*n* = 77).

#### 3.3.1. Feed Withdrawal

For feed withdrawal before catching and loading, 19% of broiler and 69% of laying hen farmers withdraw feed earlier than 12 h before slaughter, thus violating the limit of the 12 h stipulated in EU legislation [12] and potentially causing hunger and intestinal cell breakdown in poultry [7,36]. Scientific recommendations suggest fasting six to eight hours before the scheduled loading time [7]; this recommendation was met by 54% of broiler and 15% of laying hen farmers. Furthermore, laying hen farmers (13.3 ± 6.0 h) (*n* = 26/51) withdrew feed earlier than the broiler farmers did (7.6 ± 3.1 h) (*n* = 59/77) (Figure 1).

#### 3.3.2. Extra Selection Before Catching and Loading

More broiler than laying hen farmers performed a pre-transport selection round shortly before the catching and loading event to remove birds unfit for transport (39% vs. 25%; *p* = 0.06) (Figure 1). These broiler and laying hen farmers selected birds unfit for transport based on the same selection criteria during the production cycle, such as stunted growth, leg problems/lameness, feather pecking, *E. coli*, and huddled chickens. One broiler farmer only removed the dead birds during this extra selection; ill and injured birds were not selected. Animals that are already sick, weak, and injured are more vulnerable to increased suffering and have a higher chance of dying during transport because of the interactions with unfamiliar animals, difficulty maintaining stability during transit, fatigue, lack of access to feed and water, and dealing with extreme thermal environments [33,37,38]. The EU legislation [8] and scientific research [7] stipulate that unfit animals shall not be transported if they meet the following criteria: (1) inability to move independently without pain or walk unassisted (severe lameness: unable to stand or walk more than a few steps), (2) evident signs of illness, (3) broken bones (legs and wings) and dislocations, (4) presence of a severe open wound and prolapse, (5) cachexia and emaciation, (6) wet plumage in low effective temperatures (except for ducks and geese), and (7) poor feather cover in end-of-lay hens [7]. In the survey, not all of these criteria were mentioned as options, but a text box was provided if the farmers wished to add their own criteria.

EU legislation and scientific research both recommend a selection before catching and loading [7,8], more specifically within 12 h [7]. Not all poultry farmers performed this pre-transport selection within 12 h before catching and loading: a minority of the broiler (19%) and laying hen (7%) farmers performed this selection too early (>12 h). The survey responses indicate that, on average, broiler farmers (6.8 ± 7.0 h) (*n* = 37/77) started earlier with the additional selection just before catching and loading than laying hen farmers (5.1 ± 5.9 h) (*n* = 14/51) and within 12 h, but these results are not significant (*p* = 0.29). The additional selection lasted on average for about 1.2 ± 0.9 h for laying hen flocks and about 0.9 ± 0.8 h for broiler flocks (*p* = 0.85).

In summary, most chicken farmers did not perform an additional pre-transport selection before catching, and specific criteria such as the presence of a severe open wound and prolapse, cachexia and emaciation, wet plumage in low effective temperatures, and poor feather cover in end-of-lay hens were not mentioned as selection criteria despite the opportunity to do so.

#### 3.3.3. Water Withdrawal

Regarding the withdrawal of water as preparation prior to catching, laying hen farmers (*n* = 14/51) withdrew water earlier (47.9 ± 51.1 min before the onset of catching) than the broiler farmers (*n* = 48/77, 20.6 ± 23.3 min before the onset of catching). Research indicates that water should be provided close to catching, but this is not mentioned in the EU legislation. Water provision prevents thirst, which is known to negatively affect the animals’ fitness and increase the risk of severe dehydration [7,8,20,39]. Water was provided by 71% of broiler farmers (between 0 and 15 min before catching and loading), and in contrast, only 27% of laying hen farmers (between 15 and 120 min before catching and loading).

#### 3.3.4. Other Preparations Specific for Laying Hens

Among the 36 laying hen farmers without a cage system, seven removed the litter (average: 27.6 ± 21.4 h); another eight closed the area under the aviary system with fences (average: 9.1 ± 11.6 h), and 14 closed the laying nests (average: 4.9 ± 5.8 h) (Figure 1). These steps simplify the catching process, reduce carrying distances, improve container transport, and reduce slippage, resulting in less stressed chickens and improved working conditions [1,7,40].

In summary, better planning based on the expected slaughter time is needed. Clear guidelines and agreements between the poultry farmer, catching team, transporter, and slaughterhouse are essential, and taking into account differences between layers and broilers, especially when catching birds in an aviary system (during the dark period because laying hens are easier to catch in the dark). The event planner should specify actions aligned with the slaughter schedule.

### 3.4. Catching and Crating for Transport

Options for lighting during the catching of laying hens were presented as completely dark, dimming lights, and no change in the lighting schedules (e.g., dark outside; thus, no change in the lightning schedules was required). The option of completely dark being selected occurred the most (66%). There was no difference in completely dark and dimmed lights for broiler farmers as changes in the lighting schedules (Table 3). No poultry farmers indicated using blue light in the stable, which calms chickens and provides the catchers with more visibility during the catching process, thus minimizing the risk of accidents [1,39,41]. For layers, catching always began in the evening (from six to 12 p.m.) (Table 3). In contrast, the starting time for catching broilers showed a wide range from the early morning to the late evening. For layers, catching and loading typically ended late in the evening or early in the morning, whereas, for broilers, it extended from late evening until noon (Table 3). Catching, crating, and loading occurred predominantly in the dark as the animals are calmer, less stressed, and easier to catch [1,20,42,43,44]. Furthermore, 20% of the broiler farmers and 10% of the laying hen farmers expressed a preference for daytime loading, with broilers reportedly being calmer during the day compared to laying hens. To facilitate daytime catching, the use of curtains in front of the opening of the stable is recommended to reduce sunlight and help keep chickens calm [18].

Except for one broiler flock caught mechanically with the Chicken Cat Harvester (Antwerp, Belgium), all layers and broilers were caught manually, by one leg (52% and 82%, respectively) or two legs (48% and 18%, respectively with *p* = 0.001). Although, catching by both legs is recommended as the best practice [7,41,45]. The mean and median for laying hen and broiler farms is for a catcher to carry and crate five chickens at a time. Remarkably, on 4% of the broiler farms, more than nine chickens were carried and crated at a time (Table 3). On average, the catching team for laying hens consisted of more catchers than for the broilers (22 vs. 8; *p* < 0.0001). This is explained by the increased difficulty of catching layers due to the different housing systems compared to broilers [1]. For most catching events, the farmers stated that they were present from the beginning until the end of the catching and loading, and a small percentage was present only at the beginning or the end (Table 3). During catching and loading, poultry farmers were mainly supervised, and some also gave instructions (layers: *p* = 0.0005 and broilers: *p* < 0.0001) or helped with the catching (*p* < 0.0001). Poultry farmers stated that they prioritized the handling quality by the catchers of the chickens compared to the costs associated with the catching and loading process (layers: *p* = 0.008 and broilers: *p* < 0.0001). Specific for laying hen farmers, the respect of the catchers for the infrastructure of the stable was more important than the costs (*p* = 0.03). For the broiler farmers, the catchers’ handling quality was more important than the catcher’s respect for the infrastructure of the stable (*p* < 0.0001). In contrast, layer farmers prioritized the costs and the handling of the infrastructure more than the broiler farmers (Table 3). Furthermore, the catchers’ attitude impacts the animals’ handling quality [46,47].

More broiler farmers than laying hen farmers mentioned that the type of container/crate influenced animal welfare (*p* = 0.001) (Figure 2). According to [48], the design of the container/crate plays an important role, as improper crating can cause harm [40]. Modular systems placed near the chickens have been shown to reduce DOA rates [44]. The poultry farmers believe that catching poultry is more physically exhausting than mentally stressful for catchers (layers: 69% vs. 42% and broilers: 73% vs. 42%; both *p* < 0.0001). Several studies confirm that manual catching is physically demanding, mostly experienced as unpleasant by the catching teams, and the welfare of the birds may be further compromised by fatigue among the catchers [18,45]. Bottlenecks during catching and loading, such as excessive noise (layers: 28 ± 20% and broilers: 25 ± 19%), injuries due to catching (layers: 20 ± 19% and broilers: 23 ± 23%), and an uneven distribution of chickens in containers (layers: 22 ± 20% and broilers: 28 ± 29%) were present to a lesser extent according to the poultry farmers. Communication with the catching team was scored as rather good by both laying hen and broiler farmers 84 ± 19% vs. 73 ± 26%; *p* = 0.14). Surveyed poultry farmers reported positive outcomes from the catching process, including good communication with catchers, minimal noise, reduced injuries to the animals, and an even distribution of chickens in containers, all contributing to a smoother progression of the catching and loading. The majority of both layer and broiler farmers paid the catching team per chicken in comparison with payment per stable or payment per hour (81% vs. 0% vs. 13% and 76% vs. 2% vs. 19%; both *p* < 0.0001), with an average price of EUR 0.19 ± 0.06 and EUR 0.034 ± 0.003 per laying hen and broiler chicken, respectively. For a flock of 20,000 chickens, this amounts to EUR 3800 for layers and EUR 680 for broilers. In Belgium, the poultry farmer hires a catching team from a specialized company to catch their poultry or asks for help from family and friends. Teams from a catching company are trained in good practices to catch and crate poultry [49].

### 3.5. Farmers’ Opinions About Catching Methods

Figure 3 shows the farmers’ preference ranking for 10 (broiler) or 9 (layers) different catching methods according to cost and time efficiency, catcher well-being, and bird welfare. Catching and holding chickens inverted was preferred to upright catching for cost and time efficiency, catcher well-being, and bird welfare. Furthermore, this is a common method used by farm staff [7]. Moreover, there was no significant difference in preference for catching chickens by one or two legs with regard to cost and time efficiency, catcher well-being, and bird welfare. These opinions are not in line with legislation, best practices, or scientific evidence. Regulation 1/2005 [8] prohibits lifting or dragging animals by their legs or causing unnecessary pain. Best practices recommend catching poultry with two legs while supporting the body, with a maximum of three birds at once [7,43,50]. Furthermore, maximizing the weight per lift by 2 kg in laying hens results in less physical strain for the catcher [51].

In addition, according to the surveyed broiler farmers, carrying one chicken upright in contrast to holding more than three birds inverted per hand is considered more animal-friendly but less efficient. Specific to upright catching, no significant difference was found in the farmers’ preferences between catching one or two chickens. Advice has been given to avoid inverted carrying of chickens, as it can cause more injuries (e.g., dislocated joints, leg or wing fractures, and bruises) and stimulate more wing flapping compared to an upright position [7]. Upright catching could improve animal welfare, with a reduction in chicken stress and avoidance of discomfort (obstruction of breathing due to the absence of a diaphragm) by not being held upside-down [39,52]. Furthermore, upright catching results in shorter handling durations than catching by the legs of four or more chickens because only one or two chickens are caught per time [53]. Upright catching reduces stress and injuries but requires 70% more labor and is more exhausting, according to the catchers [22,51,52]; however, no difference was found between inverted and upright catching in physical strain by an ergonomist for laying hen catchers [51]. In addition, with the same number of catchers, the total catching process of upright catching takes longer compared to inverted catching [51,52].

Only one broiler farmer used mechanical catching, which, according to the surveyed broiler farmers, is better for the catcher’s well-being compared to manual catching but worse for animal welfare and cost and time efficiency. Some scientific studies indicate that mechanical catching reduces stress (limited contact between human and animal) [23] and injuries [23,25,54] if the catching machine is operated correctly with trained personnel [7], whereas other studies report more stress and injuries with mechanical catching [24,55,56]. Mechanical catching has limitations such as extra costs (purchase, transport, and cleaning), elevated biosecurity risks (difficult to clean the machine and can result in a higher chance of contamination), and the requirement for stables to be accessible for large machinery [54,57].

In summary, inverted catching is common in Flanders, with more laying hens typically caught by one leg compared to broilers in this study, though catching by both legs is recommended as best practice [7,43,50]. Upright catching can improve animal welfare but is more time-consuming and costly, according to the surveyed poultry farmers, and is perceived as more tiring by the catchers [51]. Mechanical catching has potential benefits for catchers, according to the surveyed poultry farmers and animal welfare [23,25,54], but it is not widely used in Flanders due to its limitations.

## 4. Conclusions

In conclusion, educating and sensitizing poultry farmers and their staff about culling poorly conditioned animals is vital to minimize animal suffering during the pre-transport phase. Establishing and using guidelines for selecting chickens throughout the production cycle is essential. Only a minority of the poultry farmers performed an additional selection within 12 h before catching. Raising awareness about performing this additional selection is essential because it can prevent the suffering of unfit birds during transport. Furthermore, some poultry farmers withdraw feed and water too early, emphasizing a correct timeframe for the withdrawal of feed and water as essential to avoid suffering by birds (between 8–12 h before slaughter). While upright catching improves animal welfare according to scientific research, it is not practiced by the farmers in our survey because it is deemed inefficient and negatively affects the catcher’s welfare. Despite the fact that catching chickens by two legs is recommended as best practice, more laying hens were caught by one leg compared to broilers. Mechanical catching could benefit catchers but has other associated (hygienic, structural) risks and is currently uncommon in Flanders. Overall, effective communication and respect for animal welfare are essential to reduce noise and injury to both animals and catchers and to maximize the efficiency of catching and crating.

## Figures and Tables

**Figure 1 animals-14-03241-f001:**
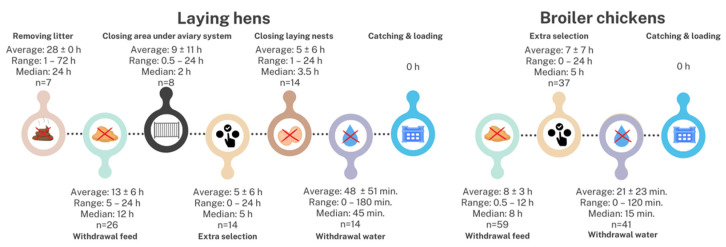
Timeline on how much time before catching the different preparations took for laying hen (*n* = 51) and broiler (*n* = 77) farmers with timeframes indicated by the surveyed poultry farmers. Specifics for the extra selection: 57 laying hen farmers and 94 broiler farmers answered the question.

**Figure 2 animals-14-03241-f002:**
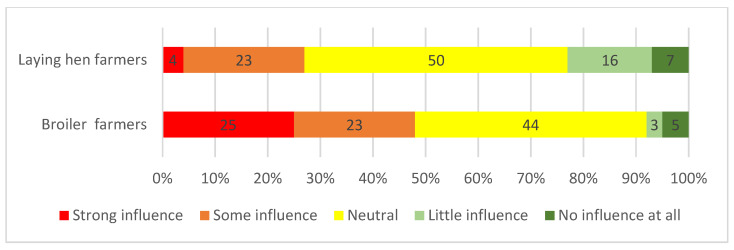
Opinion on the influence of the container/crate on animal welfare by laying hen and broiler farmers.

**Figure 3 animals-14-03241-f003:**
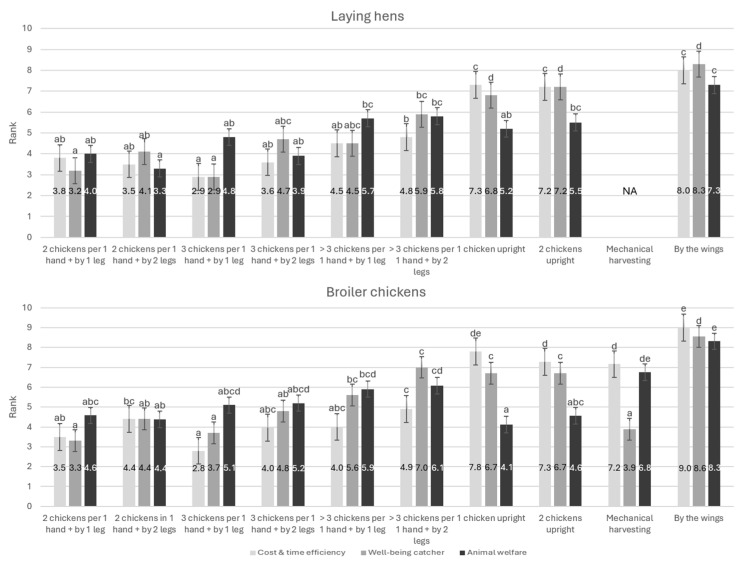
The preference of 9 or 10 different catching methods based on three categories, cost and time efficiency, the well-being of the catcher, and animal welfare by broiler (*n* = 52) and laying hen (*n* = 38) farmers ranked from 1 (most preferred) to 9 or 10, respectively (least preferred). Significant differences between catching methods per category are indicated with a, b, c, d, and e superscripts, with NA = not applicable.

**Table 1 animals-14-03241-t001:** Characteristics of the laying hen and broiler farmers/farms who filled out at least a part of the survey.

Characteristics	Laying Hen Farmers	Broiler Farmers	*p*-Value
Age (%)	*n* = 80	*n* = 133	NA
<18–25 years	0 (*n* = 0)	2 (*n* = 2)	
25–35 years	8 (*n* = 6)	17 (*n* = 22)	
35–45 years	14 (*n* = 11)	14 (*n* = 19)	
45–55 years	25 (*n* = 20)	35 (*n* = 46)	
55–65 years	50 (*n* = 40)	32 (*n* = 42)	
>65 years	3 (*n* = 3)	2 (*n* = 2)	
Gender (%)	*n* = 80	*n* = 132	NA
Male	79 (*n* = 63)	80 (*n* = 106)	
Female	21 (*n* = 17)	20 (*n* = 26)	
X	0 (*n* = 0)	0 (*n* = 0)	
Highest education level (%)	*n* = 74	*n* = 131	NA
High school degree	70 (*n* = 52)	68 (*n* = 89)	
University college degree	20 (*n* = 15)	25 (*n* = 33	
University degree	10 (*n* = 7)	7 (*n* = 9)	
Location of farms (Flemish provinces) (%)	*n* = 80	*n* = 134	
Antwerp	39 (*n* = 31) ^a^	37 (*n* = 49) ^a^	0.96
Limburg	19 (*n* = 15) ^ab^	11 (*n* = 15) ^bc^	0.29
East-Flanders	10 (*n* = 8) ^b^	16 (*n* = 21) ^c^	0.46
Flemish Brabant	1 (*n* = 1) ^b^	2 (*n* = 3) ^b^	0.86
West-Flanders	31 (*n* = 25) ^a^	34 (*n* = 46) ^a^	0.87
Housing system (%)	*n* = 57	NA	NA
Aviary	42 (*n* = 24)		
Cage	37 (*n* = 21)		
Ground	12 (*n* = 12)		
Breed (%)	*n* = 57	*n* = 110	NA
Lohman	26 (*n* = 15)	NA	
Isa Brown	26 (*n* = 15)	NA	
Dekalb White	18 (*n* = 10)	NA	
Nova Brown	12 (*n* = 7)	NA	
Bovan Brown	14 (*n* = 8)	NA	
Roman Classic	4 (*n* = 2)	NA	
Ross 308	NA	95 (*n* = 104)	
Sasso	NA	2 (*n* = 2)	
Hubbart	NA	3 (*n* = 4)	
# of chickens in stable (%)	*n* = 57	*n* = 110	NA
<10,000	23 (*n* = 13)	10 (*n* = 11)	
10,000–20,000	33 (*n* = 19)	41 (*n* = 45)	
20,001–30,000	28 (*n* = 16)	25 (*n* = 27)	
30,001–40,000	7 (*n* = 4)	15 (*n* = 16)	
>40,000	9 (*n* = 5)	10 (*n* = 11)	

Superscripts (a, b, c) indicate pairwise significant differences within the column, with NA = not applicable.

**Table 2 animals-14-03241-t002:** Multiple choice answers (multiple answers are possible and lead to summed percentages higher than 100%) about the performer, reasons, bottlenecks, and advantages of selecting chickens during the production cycle according to laying hen and broiler farmers.

Selection	Laying Hen Farmers	Broiler Farmers	*p*-Value
Performer (% of respondents)	*n* = 66	*n* = 103	
Farmer	79 ^a^	90 ^a^	**0.04**
Farm staff	14 ^b^	4 ^c^	**0.03**
Company veterinarian	0 ^c^	15 ^b^	0.10
No one	21	29	NA
Reasons (% of respondents)	*n* = 44	*n* = 86	
Stunted growth	14 ^c^	71 ^a^	**<0.001**
Leg problems/lame birds	57 ^a^	85 ^a^	**<0.001**
Feather pecking (victim)	43 ^ab^	3 ^b^	**<0.001**
*E. coli*	32 ^abc^	26 ^c^	0.45
Huddled chickens	18 ^bc^	13 ^bc^	0.41
Bottlenecks (% of respondents)	*n* = 66	*n* = 103	
Financial	15 ^b^	47 ^a^	**<0.001**
Time	33 ^a^	35 ^a^	0.83
Insufficient knowledge	8 ^b^	7 ^b^	0.85
Advantages (% of respondents)	*n* = 66	*n* = 103	
Animal welfare	55 ^a^	74 ^a^	**0.01**
Uniformity of the animals	29 ^b^	66 ^a^	**<0.001**
Preventing of disease	49 ^ab^	63 ^ab^	0.06
Less feed waste	30 ^b^	47 ^b^	**0.04**

Significant results (*p* < 0.05) between laying hen and broiler farmers are indicated in bold, and *p*-values between 0.05 and 0.10 are underlined. Superscripts (a, b, c) indicate pairwise significant differences within the column, with NA = not applicable.

**Table 3 animals-14-03241-t003:** Lighting schedules, beginning and end of catching and loading, number of chickens per catch by the catcher, presence of the poultry farmer during catching and loading, and the task of the poultry farmer during catching and loading according to laying hen and broiler farmers.

Catching and Loading	Laying Hen Farmers	Broiler Farmers	*p*-Value
Lighting schedules (% of respondents)	*n* = 35	*n* = 60	0.99
Completely dark	66 (*n* = 23)	50 (*n* = 30)	NA
Dimming lights	26 (*n* = 9)	50 (*n* = 30)	NA
No change	9 (*n* = 3)	0 (*n* = 0)	NA
Start time (% of respondents)	*n* = 42	*n* = 65	NA
0–2 a.m.	0	17 (*n* = 11)	
3–5 a.m.	0	25 (*n* = 16)	
6–8 a.m.	0	26 (*n* = 17)	
9–12 a.m.	0	2 (*n* = 1)	
1–3 p.m.	0	0	
4–5 p.m.	0	0	
6–8 p.m.	62 (*n* = 26)	3 (*n* = 2)	
9–12 p.m.	38 (*n* = 16)	28 (*n* = 18)	
End time (% of respondents)	*n* = 42	*n* = 65	NA
0–2 a.m.	55 (*n* = 23)	12 (*n* = 8)	
3–5 a.m.	19 (*n* = 8)	9 (*n* = 6)	
6–8 a.m.	2 (*n* = 1)	28 (*n* = 18)	
9–12 a.m.	0	37 (*n* = 24)	
1–3 p.m.	0	3 (*n* = 2)	
4–5 p.m.	0	6 (*n* = 4)	
6–8 p.m.	0	0	
9–12 p.m.	24 (*n* = 10)	5 (*n* = 3)	
# of chickens per catch (% of respondents)	*n* = 42	*n* = 65	NA
2	17 (*n* = 7)	0	
3	0	0	
4	33 (*n* = 14)	25 (*n* = 16)	
5	12 (*n* = 5)	60 (*n* = 39)	
6	29 (*n* = 12)	9 (*n* = 6)	
7	0	2 (*n* = 1)	
8	5 (*n* = 2)	2 (*n* = 1)	
9	2 (*n* = 1)	0	
10	2 (*n* = 1)	2 (*n* = 1)	
12	0	2 (*n* = 1)	
Presence of poultry farmer (% of respondents)	*n* = 43	*n* = 65	NA
Beginning	5 (*n* = 2)	9 (*n* = 6)	
End	2 (*n* = 1)	0	
Entire process	93 (*n* = 10)	91 (*n* = 59)	
Task poultry farmer (% of respondents)	*n* = 42	*n* = 65	
Supervision (observing)	72 (*n* = 31) ^a^	77 (*n* = 50) ^a^	0.57
Instructions	30 (*n* = 13) ^b^	23 (*n* = 15) ^b^	0.41
Catching	12 (*n* = 5) ^b^	18 (*n* = 12) ^b^	0.34
Important aspects (% of respondents)	*n* = 44	*n* = 66	
Economics	59 (*n* = 26) ^b^	36 (*n* = 24) ^b^	**0.02**
Respect for infrastructure	84 (*n* = 37) ^a^	42 (*n* = 28) ^b^	**<0.0001**
Handling of chickens	89 (*n* = 39) ^a^	95 (*n* = 63) ^a^	0.19

Significant *p*-values (*p* < 0.05) between laying hen and broiler farmers are indicated in bold. Superscripts (a, b) indicate pairwise significant differences within the column, with NA = not applicable.

## Data Availability

Data are contained within the article and Appendix A.

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
