# Peer review of "Survey of Flemish Poultry Farmers on How Birds Fit for Transport to the Slaughterhouse Are Selected, Caught, and Crated and Their Opinions Regarding the Pre-Transport Process"

_animals, 2024, doi:10.3390/ani14223241_

Round 1
Reviewer 1 Report
Comments and Suggestions for Authors
General Comments
The manuscript describes a straightforward internet-based survey of the way flocks of both egg layers and meat chickens are managed on the farm immediately prior to clean out of the barns at end of production cycle. The submission has 11 authors, which seems inflated in proportion to the complexity of the work. My fear of author inflation is bounded by the author contribution declaration at line 460. No author is identified with the chicken meat industry. End of lay hens are unlikely to have meat salvaged at the same processing facilities as broiler meat chickens. The difference between broiler meat and end of layer meat is dramatic and a comment on this may be helpful, first paragraph introduction.
The essence of the manuscript is about on farm poultry farmers behaviour in preparing birds for removal and transport. This paper is primarily about farmer behaviour and animal welfare is a secondary topic and perhaps not as relevant as the EU legislation a surrogate. The recommended farmer procedures are intended to assure the birds arrive at point of slaughter alive, well hydrated with minimal stress and empty lower gastrointestinal tracts. This is a goal shared by all vertically integrated poultry production systems worldwide, so the paper had broad readership.
Presumably if standardized or recognized timed events are not followed there is a financial loss at meat harvest, although this is probably contested by experts in the production systems. For the international reader, it would increase the value of the manuscript to describe the financial organizational structure of the system, specifically which contributor in the value chain takes the financial loss resulting from the downgrading of meat or increased death loss due to, for example failure to restrict feed at the recommended time. The belief that the legal obligations described actually make a difference to animal welfare and meat quality is not challenged by the authors and not necessary for the manuscript but may be worth mentioning (reviewer bias).
In contract broiler production in North America, the farmer does not own the animals, or the feed given to them. A contract signed with the complex that owns the hatchery feed mill and slaughterhouse, decides at what point in time the `counting` occurs and the farmers payment is calculated. These contracts are fairly complex because of the moral hazard inherent in the arrangement. Where the farmer pays for feed, and or the chickens there is high motivation to kill poor doing chickens as early in the production cycle as possible and results in decreased weight variability at slaughter and perhaps lower numbers of DOA. In Canada who is responsible for what aspect of getting poultry safely to slaughter is highly contested in the court (attached documents example from https://www.canlii.org/en/ca/cart/). There are several cases available on this website of death of end of lay chickens trying to get them to slaughter alive in Canada.
In Canada with supply management in poultry it is more common for farmer co-operatives to own the hatchery, feed mill and slaughterhouse and farmers to own the chicks at time of placement. So, financial inducements for assuring meat product quality can differ. Standard broiler production methods are encouraged by national organizations, the “raised right” logo of the Chicken Farmers of Canada, www.chicken.ca. If there is a Flanders based public promotion organization for animal welfare that pushes production to some uniform standard for animal welfare that information is relevant in the introduction as an existing mimetic force in Flanders to encourage animal care.
Payment to the farmer usually reflects quality aspects of broiler chicken both as a proportion of chicks set, and the resulting homogeneity of carcass weight at slaughter with minimal downgrading. In cull layers in North America the efficiency of production of broiler chicken meat can make it uneconomical to salvage chicken meat from end of lay hens. With the extreme low value of cull hens, there can be little financial incentive available to motivate the farmer to put effort into preparing cull hens for transport.
If the farmer has ownership of the poultry until the end of slaughter provides a different behavioural paradigm than if the farmer is paid based on live poultry loaded at farm gate. As this is a human behaviour study the manuscript should describe the financial arrangement within in these two very different production models.
Specific Comments
Line 220 concept of `thinning` I assume there two market weights that a broiler grow cohort can be marketed at. This does not occur in North America where all broilers meeting market weight are loaded and slaughtered at the same time. For Kentucky Fried Chicken – where they prevent the greasy taste of fried chicken by pressure cooking in plant oil, pieces of chicken that are 100% free of chicken fat this requires very uniform chicken at about 34-35 days old. The K-cut is an automated machine that cuts weight matched, identical chicken carcasses, in 9 pieces, specifically for KFC premium contracts. Most NA broiler markets take significantly larger birds than KFC up to 41 days old. If this thinning involves selling premium lean chicken to a specific market that information would increase the readability of the paper.
Line 222 – describe the difference between white and brown laying hens that would make one more susceptible to capture loading and transport stress.
Line 389 – farmers paid the catching team – I find this novel. In North America chicken catching teams are generally a specialized business and contracted by the slaughterhouse. Please clarify and add a note on the specialization of poultry catching teams; are the individuals accumulated ad hoc or are there specialized groups?
Figure 3 – please make this image 40% bigger; as at this size, the font is almost unreadable for the average reader. As an electronic journal there is no penalty for larger figures.
Line 459 – Does the phrase 5. Patents serve any purpose?

Author Response
Please find the Author's Notes to the Reviewer attached.

Reviewer 2 Report
Comments and Suggestions for Authors
GENERAL COMMENTS: The article addresses an important issue regarding the views of laying hen and broiler producers on the pre-transport stages. As the authors point out, even today, there are gaps in information about good welfare practices for these animals during transport, and there is a need for training and appropriate communication to understand how to transport them in a way that reduces injury and promotes good welfare. Overall, the article is well presented. But I'm afraid that the way the research was carried out, the local interest of the study (in this case, in a region of Belgium) and the scope of the conclusions would not justify a short communication rather than a full-length article. I'll leave it to the associate editor to check the relevance of this analysis. I also found the questionnaire too long, with many questions that were too specific, which may have partly justified the low number of responses from farmers. Finally, I'd like to make a few general comments, one by one, as the manuscript is very well written.
INTRODUCTION: I found it very long and tiring. I would ask the authors to reduce it to one or one and a half pages without lowering the quality of the explanation and the central problem of the research.
RESULTS AND DISCUSSION: Although very well discussed in the literature, this point lacked statistical results. In other words, only Table 2 and Figure 3 show the results. The rest is qualitative and exploratory, without depth. I want the authors to think about how they can explain the analyses they have carried out in more detail (or analyse them with other models) so that they do not just get bogged down in the numerical results of the answers. Finally, the Figure 3 is too small to visualise the details.
Author Response

(The authors gave the same response as above.)

Reviewer 3 Report
Comments and Suggestions for Authors
Both logical flow and language need a great deal of editing. The welfare of commercial poultry is very important as poor livestock welfare can affect food safety. References need checking.
1: What is the main question addressed by the research?
Is the handling of discarded layers and broilers adequate to meet legislated poultry welfare criteria in the country under study?
2: Do you consider the topic original or relevant to the field?
Yes very relevant.
3: Does it address a specific gap in the field? Please also explain why this is/ Yes. It draws attention to a lack of knowledge in regard to poultry welfare and legal consequences for poultry owners in a whole country. It is SHOCKING. This is an EU country that is supposed to be civilized. 4: What does it add to the subject area compared with other publishedmaterial? Knowledge about criminal negligence ( it highlights that poultry owners in the country are breaking EU regulations ) in regard to poultry management and welfare. 5: What specific improvements should the authors consider regarding the
methodology? The methodology is adequate. Could be better described. Editing is very poor. Sample size is acceptable - representing all poultry producers in a country. 6: What further controls should be considered? Inclusion and exclusion criteria are not stated in the publication 7: Are the conclusions consistent with the evidence and arguments presented and do they address the main question posed? Yes 8: Please also explain why this The main question was whether humane handling of cull poultry was in line with published criteria. The answer was "no". 9: is/is not the case; Are the references appropriate? Yes. But they need editing. There may be too many for the journal chosen. 10: Any additional comments on the tables and figures. Very poorly designed. Layout not acceptable and not standard. One bar chart appears to be cut and pasted from a thesis. Comments on the Quality of English Language
The logical flow and language editing is VERY poor. The topic is good and the welfare implications are important.
Author Response

(The authors gave the same response as above.)

Round 2
Reviewer 2 Report
Comments and Suggestions for Authors
Thanks for the corrected version. It's excellent.
Author Response
Thank you very much, we are glad to hear that the corrected version is excellent.
Reviewer 3 Report
Comments and Suggestions for Authors
The editing is excellent. The only comment is that it is still too long and could be shortened considerably.
Author Response
Thank you very much, we are glad to hear that the edited version is excellent. We have carefully reviewed the manuscript to reduce repetition, and shorten and streamline the text, to ensure key information remains intact. Data already presented in tables has been removed from the main text to avoid unnecessary repetition. We hope this is concise enough because otherwise, we are afraid that very important information about the data of this research would get lost. Furthermore, we updated the tables and the figures for visibility.